# Efficient D-π-π-A-Type Dye Sensitizer Based on a Benzothiadiazole Moiety: A Computational Study

**DOI:** 10.3390/molecules28135185

**Published:** 2023-07-03

**Authors:** Fatma M. Mustafa, Mahmoud K. Abdel-Latif, Ahmed A. Abdel-Khalek, Oliver Kühn

**Affiliations:** 1Chemistry Department, Faculty of Science, Beni-Suef University, Beni-Suef City 62521, Egypt; dr_fatma238@yahoo.com (F.M.M.); m_kkhedr@yahoo.com (M.K.A.-L.); ahmedakhalek41@gmail.com (A.A.A.-K.); 2Chemistry Department, Collage of Science, United Arab Emirates University, Al-Ain 15551, United Arab Emirates; 3Institute of Physics, University of Rostock, Albert-Einstein-Str. 23–24, D-18059 Rostock, Germany

**Keywords:** DSSC, TDDFT, power conversion efficiency, diphenylamine

## Abstract

The design of highly efficient sensitizers is one of the most significant areas in dye-sensitized solar cell (DSSC) research. We studied a series of benzothiadiazole-based D-π-π-A organic dyes, putting emphasis on the influence of the donor moiety on the DSSC’s efficiency. Using (linear-response time-dependent) density functional theory ((TD)DFT)) with the CAM-B3LYP functional, different donor groups were characterized in terms of electronic absorption spectra and key photovoltaic parameters. As a reference, a dye was considered that had a benzothiadiazole fragment linked via thiophene rings to a diphenylamine donor and a cyanoacrylic-acid acceptor. The different systems were first studied in terms of individual performance parameters, which eventually aggregated into power conversion efficiency. Only the amino-substituted species showed a modest increase, whereas the dimethylamino case showed a decrease.

## 1. Introduction

Solar cells are among the most promising devices for clean energy generation. Considering the various design strategies, dye-sensitized solar cells (DSSCs) have attracted substantial attention over the past decades due to their inexpensive materials, simple fabrication process, and high power conversion efficiency in comparison with conventional high-cost silicon solar cells [1,2]. Generally, DSSCs consist of the following components: a dye-sensitized semiconductor electrode (the working electrode or photoanode); a redox electrolyte; a counter electrode; and a photosensitizer or a monolayer of dye molecules. The monolayer of dye molecules adsorbed on the semiconductor surface is responsible for light absorption in the device and consists of three basic parts, an electron-donating group (D), a π-spacer (π), and an electron acceptor (A). In DSSCs, light is absorbed by the sensitizing dye, which is anchored to a semiconducting mesoporous TiO_2_ film. Electrons are injected from the excited state of the sensitizer into the conduction band of the TiO_2_ electrode, eventually leading to an electric current. Furthermore, the redox electrolyte (typically $I_{3}^{-}/I^{-}$) regenerates the oxidized sensitizer to provide efficient charge separation. Generally, dye sensitizers covalently bind to the surface of TiO_2_, which leads to an electronic connection between them that facilitates an efficient electron injection process [3]. To enhance the DSSCs’ effectiveness, significant attention has been given to the modification of the photosensitizer. It should meet specific requirements regarding its optoelectronic properties, including the region of the electromagnetic spectrum, absorption coefficient, and band alignment [1]. 

Generally, organic photosensitizers (metal-free dyes) have attracted a lot of attention due to their ability to harvest a considerable portion of the solar spectrum with high molar absorption coefficients, from the ultraviolet to near-IR regions, which results in an increase in power conversion efficiency (PCE) values. In addition, metal-free organic dyes are a promising DSSC material due to their non-toxicity, high flexibility regarding the molecular structure, tunable absorption properties, and high molar extinction coefficient compared with metal (Ru and Zn)-based dyes [4]. It was reported that the PCE for metal-based DSSCs reached up to 13% for Zn porphyrin [5] and 11.5% for a polypyridyl ruthenium complex [6]. The reported PCE for metal-free organic dyes of donor-π-acceptor (D-π-A) frameworks bound to the surface of TiO_2_ reached up to 13% [7].

In D-π-A systems, the π-bridges facilitate the charge separation between donor and acceptor and inhibit competing charge recombination. The performance of photovoltaic cells based on organic dyes can be enhanced by choosing suitable groups within the D-π-A molecular architecture [8,9]. For instance, increasing the electron-donating strength of the donor or the electron-withdrawing strength of the acceptor can improve the performance of DSSCs [10]. Typical organic dyes with a relatively high performance include those with triphenylamine [9], carbazole [11,12], indoline [13], quinoline [14], coumarin [15], and zinc phthalocyanines [16] as electron-donating groups. On the other hand, 2-cyanoacrylic acid is the most commonly used electron acceptor, although other acceptors, like bipolar diketopyrrolopyrrole-based ones [17,18], have been theoretically studied. Phenyl, thiophene, furan, and their derivatives are the moieties most often used as π-bridges [19]. Moreover, several DSSCs with dye sensitizers based on a benzothiadiazole moiety have been reported, with efficiencies ranging from 8 to 10% [20,21,22]. 

In this work, we investigated the effect of the donor group on the optoelectronic properties of organic dyes with a D-π-π-A structure using computational chemistry. As a reference structure (**D0**), we considered the dye synthesized and characterized in Ref. [23]. It has a benzothiadiazole fragment placed between two thiophene rings as a bridge that connects the diphenylamine donor and cyanoacrylic-acid acceptor (see Figure 1). Phenylamine donors are common in DSSCs (see, e.g., the studies of the influence of the type of π-spacer in Refs. [9,24,25] or of donor properties in Ref. [26]); we considered five substitutions (**D1**–**D5**), as given in Figure 1 While the modifications were modest, it was shown that the effect on the relevant DSSC parameters was substantial. 

In what follows, we start Section 2.1 and Section 2.2 with a discussion of the geometries and the electrostatic potential maps, respectively, obtained using DFT. In Section 2.3, a frontier orbital analysis is presented. In Section 2.4, the optical properties are discussed; in Section 2.5, the electron transfer is characterized. Section 3 presents a discussion of our findings, and the computational methods are summarized in Section 4.

## 2. Results

### 2.1. Geometries

Optimized ground-state geometries were obtained and analyzed with respect to changes due to the substitutions. Overall, the bridge and acceptor moieties were essentially planar, facilitating efficient charge delocalization. This held in particular for **D4** and **D5**. For **D0** to **D2**, the thiophene unit next to the donor was twisted by about 13 degrees with respect to the benzothiadiazole unit. For **D3**, this dihedral twist was about 6 degrees. In order to better characterize the structural changes, we aligned the **D0** structure with the **D0**-type parts of **D1** to **D5** using the Kabsch algorithm to minimize the RMSD. The smallest residual RMSDs were found for **D1** (0.091 Å) and **D2** (0.116 Å); the ones for **D3** to **D5** were much larger (0.471, 0.489, and 0.485 Å). In Figure 2, overlaid structures are given to visualize these moderate differences.

### 2.2. Electrostatic Potentials

The electrostatic potential maps (EPMs) of all studied dyes at the optimized geometries in the electronic ground state are shown in Figure 3. The appearance of negative electrostatic potentials (red) around the oxygen and nitrogen atoms of the anchoring acceptor group evidenced its negative partial charge. These sites were potentially relevant for electrophilic attacks of the electrolyte, a finding that was in accord with previous studies on **D0** in Refs. [24,25]. Positive electrostatic potentials (blue) were found in the donor moieties, particularly for the NH_2_ group in **D3**. We noticed that upon photoexcitation and oxidation, the electron density was removed from the donor site, making it even more attractive for the reduced redox couple to facilitate dye regeneration (for instance, Ref. [26]). Overall, the change of the donor moiety did not have a noticeable influence on the geometry and EPM of the remaining dye.

### 2.3. Frontier Molecular Orbitals and Chemical Reactivity 

The HOMO and LUMO of a dye sensitizer must show suitable energetic positions to match the redox potential of the electrolyte and the conduction band of TiO_2_. In other words, the dye’s E_LUMO_ must be above the semiconductor’s conduction band edge, indicating that electrons can easily be injected from the excited dyes into the conduction band. At the same time, the E_HOMO_ of the dye must be below the redox potential of the electrolyte $(I_{3}^{-}/I^{-}$) couple to facilitate the regeneration of the dyes [1]. Figure 4 shows the HOMO and LUMO of all studied dyes. The electron distribution in the HOMO was mostly located on the electron donor and π-spacer, while the LUMO was mostly located on the π-conjugated moiety and the electron acceptor group in the anchoring unit. The latter behavior especially represents an ideal spatial arrangement of the molecular orbitals for DSSC applications. The ionization potential (IP = −E_HOMO_), electron affinity (EA = −E_LUMO_), chemical potential (µ), global hardness (η), electrophilicity (ω), electro-accepting power (ω^+^), and electro-donating power (ω^−^) of all studied dyes are summarized in Table 1 (for definitions, see Section 4).

It was obvious that the modification of the donor units significantly affected the HOMO energy levels of the dyes, while the energy of the LUMO, having only a little amplitude in the donor moiety, was less affected. The E_HOMO_ energies of all studies dyes were below the redox potential of the electrolyte ($I_{3}^{-}/I^{-})$ couple (−4.80 eV); this favored charge regeneration, with **D5** being the most effective case. In other words, the electron-donating power of the different donor groups increased from **D0** to **D5**. Moreover, the energies, or E_LUMO_, of all dyes were above the semiconductor’s conduction band edge (−4.0 eV), a precondition for electron injections from the excited dye to the conduction band of TiO_2_. The energy levels are sketched in Figure 5. 

When inspecting the other quantities in Table 1, we observed the following trends. The studied dyes **D1** to **D5** showed a lower energy gap compared with the reference dye (**D0**); this could enhance the light-harvesting efficiency [26]. **D5** had the smallest values for IP and EA in our calculations; thus, according to Ref. [26], it should have given the best balance between the electron and hole transfers among the considered dyes. Further, the chemical hardness and chemical potential decreased from **D0** to **D5**, leading to an increase in the electrophilicity and, thus, to a potential improvement in the charge separation capability. This, in turn, was also reflected in the electron-accepting/donating powers.

### 2.4. Optical Absorption

The experimental spectrum was available for the parent compound **D0** only. It showed peaks at 541 nm, 398 nm, and 312 nm, with an intensity ratio of 1.16/0.96/1.00 [23]. CAM-B3LYP predicted the lowest transition at 534 nm (f = 1.31). Further transitions with appreciable oscillator strength (f) were located at 382 nm (f = 0.37) and 310 nm (f = 0.26). Although the ratio of oscillator strengths did not match the experiment, the agreement of spectral positions was excellent, justifying the use of the computational setup for the following analysis. This was justified because, according to Equation (8), only the properties of a single transition were compared for the different dyes. 

The absorption spectra of all considered dyes are shown in Figure 6. The calculated values of the vertical excitation energy (E_max_) and absorption wavelength (λ_max_) for the transition with the largest oscillator strength are given in Table 2, together with the light-harvesting efficiency (LHE). For comparison, the values in the gas phase were also given, which showed a clear effect due to the THF solvent. When designing dye sensitizers, one of the essential aims is to establish a system with an optical absorption that overlaps with the sun’s emission spectrum. For the considered organic dyes, this implied an increase in the absorption toward the UV–Vis and IR ranges, with a high molar extinction coefficient. Overall, all studied dye modifications (**D1** to **D5**) had a strong absorption at longer wavelengths compared with the reference dye **D0** (bathochromic shift). Interestingly, for **D4** and **D5**, this shift was large enough to move the main peak of the spectrum into the infrared range. In all cases, the lowest and strongest band was dominated by a HOMO to LUMO transition and, thus, had an ICT character (see Figure 4). There was some admixture of HOMO-1 and LUMO+1 configurations, as indicated in Table 2. The oscillator strength increased when going from **D0** to **D5**, as did the LHE. Hence, in terms of this property, the designed dyes should have yielded an improved PCE. 

### 2.5. Electron Injection, Dye Regeneration, and Open-Circuit Voltage 

A complete cycle occurred when electrons, after photoexcitation, were spontaneously injected from the LUMO into the conduction band of the TiO_2_ semiconductor and the dyes were regenerated by the redox couple into their ground state. The driving force for electron injection ΔG_inj_ was negative for all dyes, as outlined in Table 3. Thus, the conditions for electron injection were favorable for all dyes. The ΔG_inj_ values of the investigated dyes followed the order **D3** < **D0** < **D1** < **D2** < **D4** < **D5**, although we observed that, apart from **D4** and **D5**, the values were rather similar.

Dye regeneration is determined by the potential difference between the electrolyte and the oxidized dye. This should be large enough to provide a driving force for the regeneration of the ground-state dyes. At the same time, its value also influences the recombination rate between the oxidized dye and conduction band of the TiO_2_ semiconductor. Efficient dye regeneration requires ΔG_reg_ values in the range of 0.1 to 0.3 eV [26]. Inspecting Table 3, we noticed that only **D3** fell into that range, whereas **D0** to **D2** were larger and **D4** and **D5** had approximately zero driving force. 

The open-circuit voltage, V_oc_, provides an idea about the mobility and number of charge carriers across the interface. A high-value V_oc_ results in a smaller loss due to charge recombination so that a higher value improves the cell efficiency. The calculated values of V_oc_ for the studied dyes ranged from 1.55 eV to 1.77 eV (Table 3). **D2** and **D3** had the highest V_oc_ values. Again, only **D4** and **D5** had substantially different values. 

The overall performance of the different dyes could be quantified by the PCE value defined in Equation (6) and given in Table 3. As we were interested in the effect of the donor modification, the PCE values were given relative to **D0**. Inspecting Table 3, it was clear that only **D3** provided an improvement to the PCE, whereas for **D4** and **D5**, the PCE values were considerably smaller than for **D0**.

## 3. Discussion

We performed a (TD)DFT investigation into a series of novel organic sensitizers based on the D-π-π-A structure for potential application in DSSCs. These dyes were derived from a parent compound with a diphenylamine donor and were linked via a thiophene and a thiophene-fused benzothiadiazole bridge to a cyanoacrylic-acid acceptor (**D0**). The donor was modified at the para-position, yielding **D1** to **D5**, as shown in Figure 1. First, the geometries and the ground state and frontier molecular orbitals were analyzed. Electrostatic potential maps showed little influence by the substitution. The IP decreased from **D0** to **D5**, whereas the EA increased accordingly. This, also taking into account the derived quantities (cf. Equations (1)-(5)), suggested that **D5** was the best candidate among the studied dyes for DSSC applications. **D5**, as well as **D4**, also stood out in relation to optical absorption, insofar as the bathochromic shift was rather considerable. The LHE values, however, were mostly comparable among the different systems. Inspecting the electron-transfer-related properties, an injection—in terms of ΔG_inj_—was favored for all dyes; only **D3** was in the range of reasonable regeneration as far as ΔG_reg_ was concerned. Judging the performance based on the PCE, only **D3** (i.e., substitution with an amino group) gave an improvement compared with **D0**. This was interesting insofar as some of the individual characteristics suggested that **D5** should have outperformed **D0**.

There is a large body of literature on (TD)DFT studies of D-π-A systems, even if one only focusses on those systems with a cyanoacrylic-acid acceptor (e.g., [8,9,10,12,14,24,25,26,27,28,29,30,31,32,33,34,35,36]). Most of these studies lack experimental verification and, therefore, should be viewed as an in silico screening of potential sensitizer materials. The present investigation has added to this endeavor as it set its focus on the modification of the donor moiety starting from a simple diphenylamine. We found that, for the considered systems and in terms of the PCE, there was little room for improvement (up to 11%), but the performance worsened (−25%). Closest to our work is the study of Hailu et al. [26], although they used a different spacer. In fact, our dye **D4** corresponded with their **dye1**. Apart from this dimethylamino case, they considered methylphenylamino, diphenylamino, diindoline, and dicarbazole substitutions. Interestingly, they considered the first three dyes as potentially good performers (a direct comparison is not possible because their V_oc_ was not reported). This was at variance with the poor PCE performance of **D4** in the present case, although we noticed that, in terms of IP and EA, **D4** was superior to **D0** to **D3**. A major bottleneck for a direct comparison of the reported quantum chemical results was the use of different computational protocols. For instance, in Ref. [26], the ωB97XD functional was used, whereas in the present case, it was CAM-B3LYP. In passing, we noted that both functionals contained all-purpose parameters for the range separation. More accurate results could be obtained by applying optimal tuning of these parameters for the specific types of systems [37]. 

In summary, in order to advance the field of computational screening of dyes for application as sensitizers in DSSCs, it is necessary to identify and benchmark a computational protocol to be used to establish a database of these materials.

## 4. Materials and Methods

### 4.1. Characterization of Dye Molecules

The ionization potential (IP) and electron affinity (EA) of a sensitizer describe the electronic energy barrier for the creation of holes and electrons, respectively. A lower IP should promote the hole-creating ability, whereas a higher EA should enhance the electron-accepting ability of a dye. Based on these two parameters, the chemical reactivity of model dyes can be characterized by the electronic chemical potential (μ), chemical hardness (η), and electrophilicity index (ω). The electronic chemical potential is the negative of the electronegativity, which quantifies the ability of the system to attract and retain electrons. The chemical hardness describes the resistance of a dye to a change in its electronic state, e.g., by means of intra-molecular charge transfer (ICT) in a multicomponent system, as with the present D-π-π-A. 

The electrophilicity index ω encompasses both; the propensity of the electrophile to acquire an additional electronic charge (μ) and the resistance of the system to simultaneously exchange the electronic charge with the environment (η). Thus, electrophilicity represents the stabilization energy of dyes upon acquiring an additional charge. Consequently, dyes suitable for DSSCs should have a low chemical hardness and high chemical potential to increase charge separation. 

Following Parr and Yang, the electronic chemical potential, chemical hardness, and electrophilicity index are commonly expressed by the following equations [38]:(1)$$\mu =-\frac{\mathrm{IP}+\mathrm{EA}}{2}=\frac{1}{2}\left(E_{\mathrm{HOMO}}+E_{\mathrm{LUMO}}\right),$$
(2)$$\eta =\frac{\mathrm{IP}-\mathrm{EA}}{2}=\frac{1}{2}\left(E_{\mathrm{LUMO}}-E_{\mathrm{HOMO}}\right)$$
(3)$$\omega =\frac{\mu^{2}}{\eta }$$

Further, we defined the electron-accepting (ω^+^) and electron-donating (ω^−^) power as follows:(4)$$\omega^{+}=\frac{{\left(\mathrm{IP}+3\mathrm{EA}\right)}^{2}}{16\left(\mathrm{IP}-\mathrm{EA}\right)}$$
(5)$$\omega^{-}=\frac{{\left(3\mathrm{IP}+3\mathrm{EA}\right)}^{2}}{16\left(\mathrm{IP}-\mathrm{EA}\right)}$$

The parameters ω^−^ and ω^+^ quantify the ability of these dyes to withdraw or gain electron charges; for good performance, large values are desirable. The energy gap is given as $E_{g}=2\eta$.

The overall power conversion efficiency (PCE) of DSSCs is given by the photocurrent density measured as the short-circuit (J_sc_), the open-circuit photovoltage (V_oc_), the fill factor of the cell (FF), and the intensity of the incident light (P_in_), as summarized in the following expression [31]:(6)$$\mathrm{PCE}=\mathrm{FF}\;\frac{J_{\mathrm{sc}}V_{\mathrm{oc}}}{P_{\mathrm{in}}}\times 100\%\;,$$
where J_sc_ could be determined using the following equation [35,36,39]:(7)$$J_{\mathrm{sc}}={\mathrm{LHE}\;\Phi }_{\mathrm{inject}\;}\eta_{\mathrm{collect}}\;.$$

Here, LHE is the light-harvesting efficiency at a maximum wavelength, Φ_inject_ is the electron injection efficiency, and η_collect_ is the charge collection efficiency. In systems where the only difference is in the sensitizer, η_collect_ is an assumed constant. According to Equation (7), to obtain a high J_sc_, LHE and Φ_inject_ should be as large as possible. The LHE can be expressed by the following equation:(8)$$\mathrm{LHE}=1-10^{f}$$
where f is the oscillator strength of the dye related to the maximum absorption wavelength λ_max_.

The open-circuit voltage V_oc_ in Equation (6) is related to the electron injection from the excited dye to the conduction band of the semiconductor and is determined by the following equation (neglecting occupation effects as well as conduction band shifts in the semiconductor):(9)$$V_{\mathrm{oc}}=E_{\mathrm{LUMO}}-E_{\mathrm{CB}}^{{\mathrm{TiO}}_{2}}$$
where E_LUMO_ is the LUMO energy of the dye and $E_{\mathrm{CB}}^{{\mathrm{TiO}}_{2}}$ is the conduction band energy of the semiconductor (here, TiO_2_). It is difficult to accurately determine $E_{\mathrm{CB}}^{{\mathrm{TiO}}_{2}}$ because it is highly sensitive to operating conditions such as the pH of the solution. In the present study, we used $E_{\mathrm{CB}}^{{\mathrm{TiO}}_{2}}\;$= −4.0 eV, which was the experimental value corresponding with conditions where the semiconductor was in contact with aqueous redox electrolytes with a fixed pH of 7.0 [31]. Φ_inject_ is closely related to the thermodynamic driving force ΔG_inject_ of electron injections from the excited states of dye to the conduction band of TiO_2_, according to the following relation [35]:(10)$$\Phi_{\mathrm{inject}}\propto {\Delta G}_{\mathrm{inject}}=E_{\mathrm{dye}}^{*}-E_{\mathrm{CB}}^{{\mathrm{TiO}}_{2}}.$$

Here, $E_{\mathrm{dye}}^{*}$ is the oxidation potential of the excited dye at the ground-state geometry (neglecting vibrational relaxation upon excitation) following from $E_{\mathrm{dye}}^{*}=E_{\mathrm{dye}}^{\;}-\Delta E,$ with $E_{\mathrm{dye}}$ being the oxidation potential energy of the dye in the ground state, while ΔE is the vertical electronic excitation energy corresponding with λ_max_. In order to obtain more reliable results for the oxidation potential, we used the ΔSCF method instead of Koopmans theorem; that is, $E_{\mathrm{dye}}^{\;}=E_{\mathrm{GS}}-E_{\mathrm{GS}}^{+}$, with GS referring to the ground state; the vibrational relaxation effects were neglected. The dye regeneration energy ($\Delta G_{\mathrm{regen}}$) could be calculated by the following equation [24]:(11)$$\Delta G_{\mathrm{regen}}=E_{\mathrm{redox}}-E_{\mathrm{dye}}$$
where $E_{\mathrm{redox}}$ is the ground-state oxidation potential of the triiodide/iodide redox couple electrolyte redox potential (−4.80 eV) [24].

### 4.2. Computational Chemistry

Density functional theory (DFT) calculations were performed for the determination of the optimized structures of the molecules **D0**–**D5** at the CAM-B3LYP/6-31G(d) level of theory [40] using the Gaussian09 program [41]. In fact, these molecules showed multiple conformations and we chose the one previously reported for the diphenylamine donor [26]. Optimization was followed by frequency calculations to confirm the minimum structure on the potential energy surface. Linear-response time-dependent TDDFT computations were carried out to calculate the electronic absorption spectra for the 25 lowest singlet vertical excitations. The solvent environment (tetrahydrofuran) was implicitly treated using the self-consistent reaction-field-polarizable continuum model [42]. We also calculated the absorption spectrum using the B3LYP functional. B3LYP predicted the lowest and strongest transition for **D0** at 758 nm, which was at variance with the experiment and with the CAM-B3LYP results reported in Section 2.3. In view of the results reported in Ref. [26] (although with a different spacer), we repeated the calculation of the absorption spectrum of **D5** using a 6-311+G(d) basis set. We observed no noticeable change as far as the low energy absorption peak was concerned. The HOMO and LUMO energies, HOMO–LUMO energy gap, and other parameters defined in Section 4.1 were calculated at the optimized geometry.

## Figures and Tables

**Figure 1 molecules-28-05185-f001:**
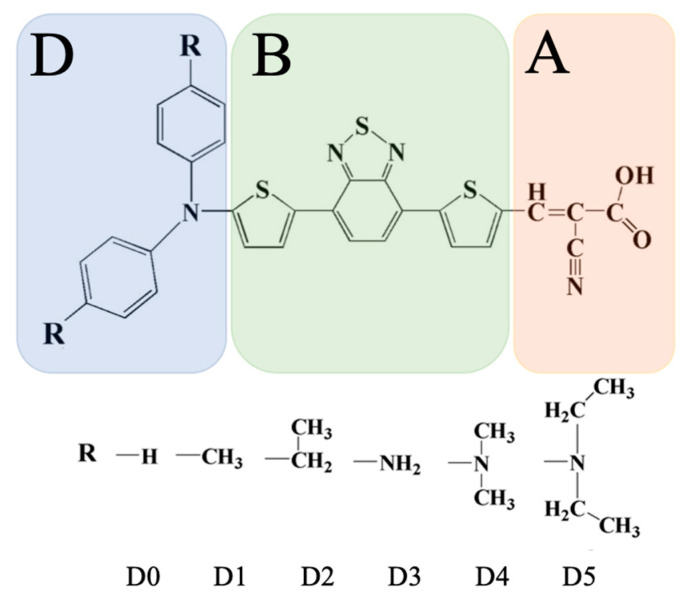
Structures of the series of dyes **D0** to **D5** with different donor moieties studied in this work. The dyes were composed of a (modified) diphenylamine donor (D); a π-π bridge (B), consisting of two thiophenes and a thiophene-fused benzothiadiazole group; and a cyanoacrylic-acid acceptor (A). Note that **D4** corresponds with **dye1** of Ref. [26], which contained a different π-spacer.

**Figure 2 molecules-28-05185-f002:**
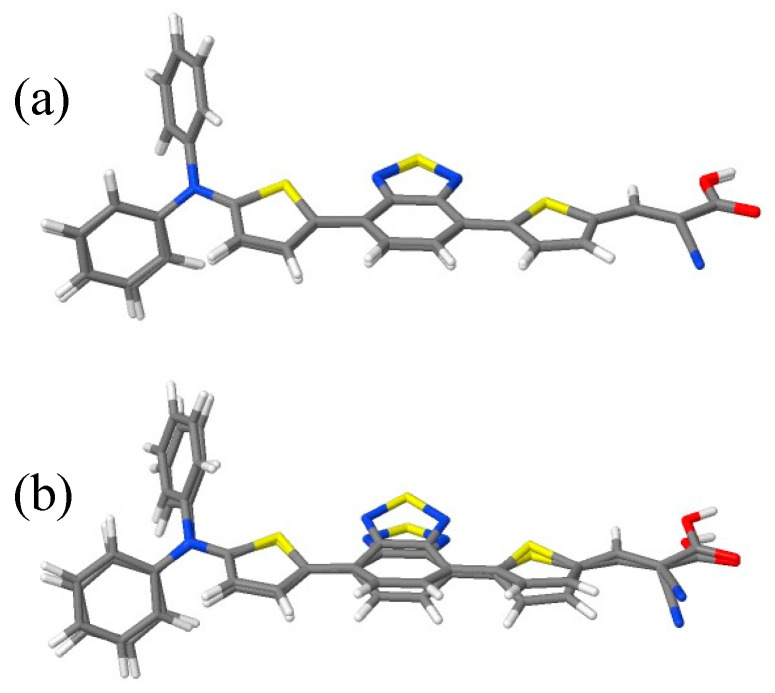
Comparison of optimized geometries for **D0** and **D1** (**a**) as well as **D0** and **D5** (**b**); only the **D0**-like parts of **D1** and **D5** are shown.

**Figure 3 molecules-28-05185-f003:**
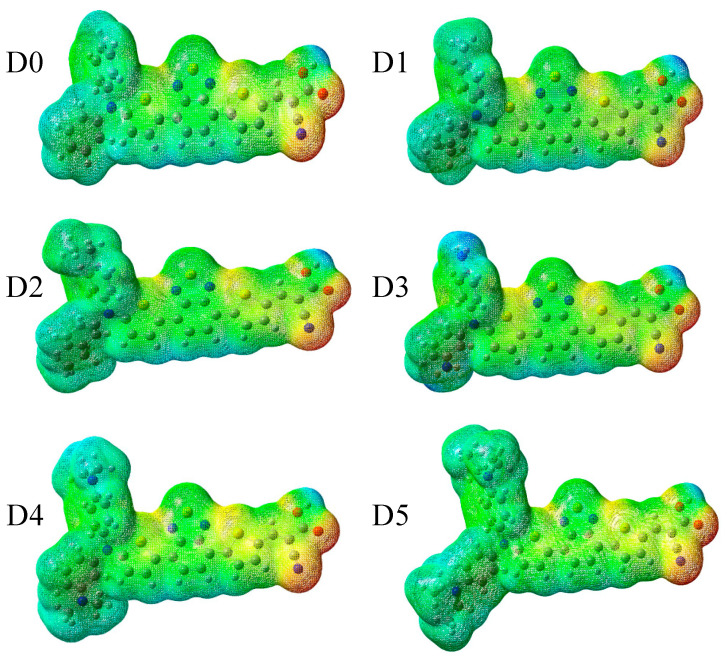
EPMs of the studied dyes (in atomic units (a.u.), which were multiplied by 27.2113 to obtain the volt unit): **D0** (colors from −0.80 (red) to 0.80 (blue) a.u.); **D1** (−0.81 to 0.81 a.u.); **D2** (−0.81 to 0.81 a.u.); **D3** (−0.80 to 0.80 a.u.); **D4** (−0.86 to 0.86 a.u.); and **D5** (−0.88 to 0.88 a.u.).

**Figure 4 molecules-28-05185-f004:**
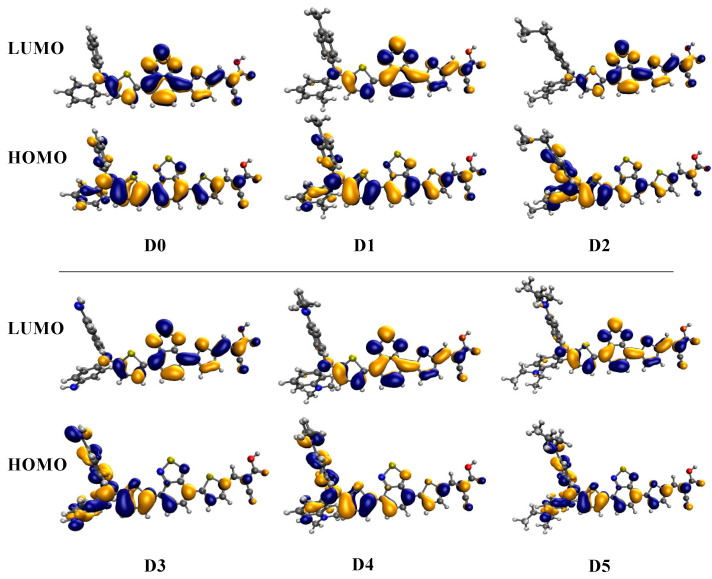
Selected frontier molecular orbitals of the considered dyes.

**Figure 5 molecules-28-05185-f005:**
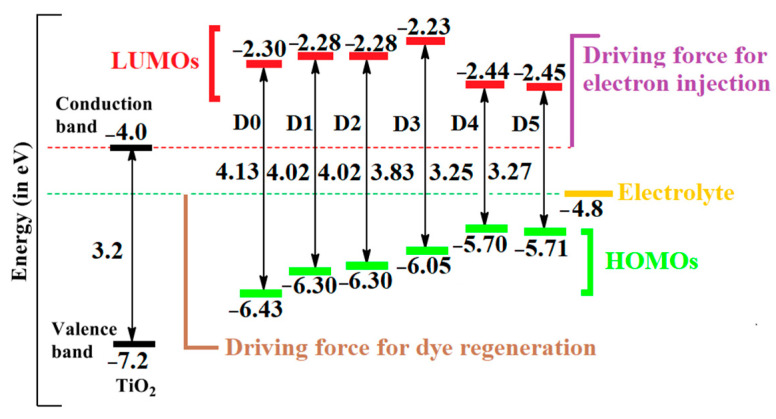
Energy-level diagrams of the studied dyes, including the conduction band of TiO_2_ and the redox potential of the $I_{3}^{-}/I^{-}$ redox couple, values of conduction, and valence bands of TiO_2_. The electrolytes were taken from Ref. [27].

**Figure 6 molecules-28-05185-f006:**
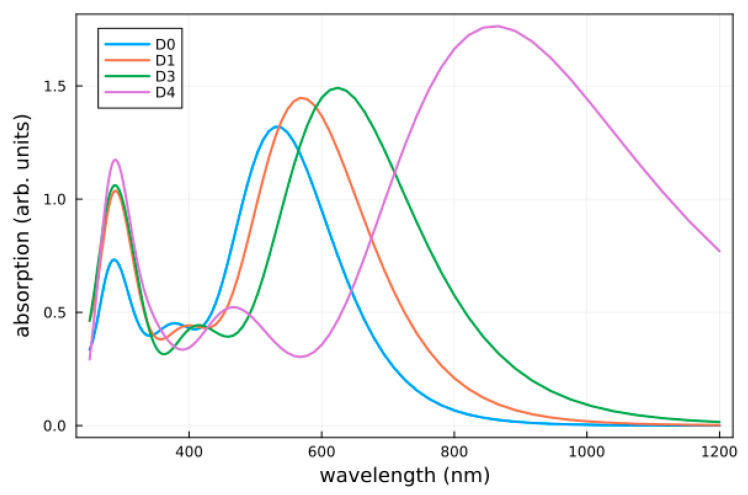
Calculated absorption spectra of the studied dyes in THF solution. The spectra of **D1** and **D2** and of **D4** and **D5** were rather similar; only **D1** and **D4** are shown. A Gaussian-type broadening of 0.1 eV was used to mimic environmental effects.

**Table 1 molecules-28-05185-t001:** Ionization potential (IP = −E_HOMO_; in parentheses are the ΔSCF values, which included the electronic relaxation effects), electron affinity (EA= −E_LUMO_), chemical potential (µ), global hardness (η), electrophilicity (ω), electron-accepting power (ω^+^), and electron-donating power (ω^−^).

Dye	IP (eV)	EA (eV)	η (eV)	μ (eV)	ω (eV)	ω^+^ (eV)	ω^−^ (eV)
**D0**	6.43 (5.45)	2.31	2.06	−4.37	9.26	2.70	10.41
**D1**	6.31 (5.31)	2.28	2.01	−4.30	9.17	2.69	10.32
**D2**	6.30 (5.31)	2.28	2.01	−4.29	9.18	2.69	10.32
**D3**	6.06 (5.07)	2.23	1.91	−4.14	8.98	2.66	10.10
**D4**	5.70 (4.75)	2.45	1.63	−4.07	10.20	3.27	11.48
**D5**	5.72 (4.76)	2.45	1.63	−4.09	10.23	3.28	11.51

**Table 2 molecules-28-05185-t002:** Vertical transition energy, wavelength, and oscillator strength of maximum absorption peak according to CAM-B3LYP TDDFT calculation. All transitions were dominated by HOMO to LUMO configurations. For **D0**–**D3**, there was some admixture of HOMO-1 to LUMO and HOMO to LUMO+1, whereas for **D4** and **D5**, it was only HOMO-1 to LUMO. In the last column, the LHE (Equation (8)) is given. In parentheses, the gas-phase values are given for comparison.

Dye	E_max_ (eV)	λ_max_ (nm)	f	LHE
**D0**	2.32 (1.88)	534.2 (658.3)	1.31 (1.04)	0.95 (0.91)
**D1**	2.17 (2.32)	570.7 (533.0)	1.43 (1.10)	0.96 (0.92)
**D2**	2.16 (2.32)	571.2 (534.0)	1.44 (1.11)	0.96 (0.92)
**D3**	1.98 (2.18)	624.3 (566.8)	1.48 (1.14)	0.97 (0.93)
**D4**	1.44 (1.74)	860.8 (709.3)	1.76 (1.32)	0.98 (0.95)
**D5**	1.44 (1.75)	857.1 (710.2)	1.75 (1.33)	0.98 (0.95)

**Table 3 molecules-28-05185-t003:** Free energies of charge injection (ΔG_inj_), dye regeneration (ΔG_reg_), and open-circuit voltage (V_oc_); all values in eV. Also given is the PCE relative to **D0** under the assumption that the FF, *η*_collect_, and *P*_in_ were the same for all dyes. The factor V_oc_
$\mathrm{LHE}\;\Phi_{\mathrm{inject}\;}$ amounted to 1.4 for **D0**.

Dye	ΔG_inj_	ΔG_reg_	V_oc_	PCE
**D0**	−0.87	0.65	1.69	100
**D1**	−0.86	0.51	1.72	101
**D2**	−0.85	0.51	1.77	100
**D3**	−0.91	0.27	1.77	111
**D4**	−0.69	−0.05	1.55	75
**D5**	−0.68	−0.04	1.55	73

## Data Availability

Data of the quantum chemical calculations are available upon request from the authors.

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
