# Peer review of "Efficient D-π-π-A-Type Dye Sensitizer Based on a Benzothiadiazole Moiety: A Computational Study"

_molecules, 2023, doi:10.3390/molecules28135185_

Round 1
Reviewer 1 Report
I have some comments which you may consider as minor revision to further improve the credibility and visibility of this manuscript. The detailed suggestions are:
1. The resolution of MEP and HOMO, LUMO is not good?
2. Please add a proper unit of each term in Table 1.
3. The UV calculation in gas phase is not present?
4. You are not discussing the value that is mentioned in Figure 2. What is this?
5. Add more recent reports in introduction part on solar cells by using DFT and TD-DFT approaches.
6. Why are you not discussing the density of states and the transition density matrix, which conforms to the electron-hole transformation in a solar cell?
7. The diagram of optimized geometries of compounds is missing, and there is no discussion about it.
8. Add more recent reports in introduction part on solar cells by using DFT and TD-DFT approaches, you are recommended to focus (doi.org/10.1016/j.comptc.2021.113242, 10.1039/D1RA04529F, doi.org/10.1016/j.comptc.2021.113454).
9. Please also mention the energy gap equation Eg = (EL-EH) in the description of molecular orbital, which is also mentioned in the above articles.
10. Why we use CAMB3LYP of optimization rather than other?
11. What is the significance of incorporating the benzothiadiazole moiety in the dye-sensitizer?
12. How to calculate the value of some terms and give its proper information source?
i) Intensity of the incident light (Pin)
ii) Charge collection efficiency
iii) Electron injection efficiency
There are many typos and grammar mistakes in the manuscript. The authors may need to correct all of these before resubmission.
Author Response
see PDF

Reviewer 2 Report
The manuscript titled "Efficient D-π-π-A Type Dye-Sensitizers Based on the Benzothiadiazole Moiety: A Computational Study" by Fatma M. Mustafa, Mahmoud K. Abdel-Latif, A.A. Abdel-Khalek, and Oliver Kühn presents a thorough investigation into the efficiency of D-pi-pi-A dyes based on benzothiadiazole (BTD) for dye-sensitized solar cells (DSSCs). The authors employed TDDFT calculations at the CAM-B3LYP level of theory to explore the performance of these dyes.
The authors made intriguing choices by utilizing a bithiophene-substituted BTD spacer, diphenylamine donor, and cyanoacrylic acid acceptor, which hold significant promise for DSSCs. In the introduction, the authors provide a comprehensive overview of the DSSC mechanism, effectively emphasizing the importance of donor variability.
The results reveal that while variations in donors exert minimal influence on the EPM, they strongly impact the HOMO-LUMO energy levels, optical absorption profile, and overall performance of DSSCs. The authors also incorporated ionization potential (IP) and electron affinity (EA) as key parameters in their studies.
Overall, this study underscores the pivotal role of theoretical calculations in advancing solar cell and clean energy applications. The advantages of theoretical calculations, such as their ease of reproducibility compared to actual synthesis and device fabrication, are important. The findings from these computational studies offer valuable insights for predicting the optimal structural architecture, facilitating subsequent experimental studies.
Based on the significance and rigor of this research, I highly recommend the publication of this article in Molecules.
Author Response
We thank the reviewer for his very positive report.
Round 2
Reviewer 1 Report
The authors did a good job while revising the manuscript. The current manuscript looks much better. As the authors have fully addressed my concerns, I think it would be publishable now. The authors should take care of typos and non-technical words in the manuscript.
The authors should take care of typos and non-technical words in the manuscript to keep everything conceptually correct.